# Clinical Outcomes and Prognostic Factors in Complex, High-Risk Indicated Procedure (CHIP) and High-Bleeding-Risk (HBR) Patients Undergoing Percutaneous Coronary Intervention with Sirolimus-Eluting Stent Implantation: 4-Year Results

**DOI:** 10.3390/jcm12165313

**Published:** 2023-08-15

**Authors:** Maciej Tyczyński, Adam Kern, Patryk Buller, Wojciech Wańha, Robert J. Gil, Jacek Bil

**Affiliations:** 1Department of Invasive Cardiology, Center of Postgraduate Medical Education, 02-508 Warsaw, Poland; maciej.tyczynski@gmail.com; 2Department of Cardiology and Internal Medicine, School of Medicine, Collegium Medicum, University of Warmia and Mazury, 10-082 Olsztyn, Poland; adamkern@mail.com; 3Department of Cardiology, Provincial Integrated Hospital, 09-400 Plock, Poland; patryk@buller.com.pl; 4Department of Cardiology and Structural Heart Diseases, Medical University of Silesia, 40-635 Katowice, Poland; wojciech.wanha@gmail.com; 5Department of Cardiology, State Medical Institute of the Ministry of Interior and Administration, 02-508 Warsaw, Poland; scorpirg@gmail.com

**Keywords:** SES, high-risk PCI, Alex Plus, major cardiovascular adverse events, high bleeding risk, in-stent restenosis

## Abstract

We aimed to characterize the performance and safety of percutaneous coronary intervention (PCI) in complex, high-risk indicated procedure (CHIP) and high-bleeding-risk (HBR) patients at a 4-year follow up. We included all consecutive patients who underwent PCI with the sirolimus-eluting coronary stent Alex Plus (Balton, Poland) between July 2015 and March 2016. We analyzed various baseline demographic and clinical characteristics, laboratory data, and clinical outcomes. We enrolled 232 patients in whom 282 stents were implanted, including 81 patients meeting the CHIP criteria and 76 patients meeting the HBR criteria. In the whole population, the mean age was 68 ± 11 years, and 23.7% were females. Most procedures were performed from radial access (83.2%) using a 6F guiding catheter (95.7%). The lesions were mostly predilated (61.6%), and postdilatation was performed in 37.9%. The device success was 99.6% (in one case, a second stent was required due to heavy calcifications). Additional stents were deployed in 39% of cases due to edge dissection (6.9%), side branch stenting (5.2%), or diffuse disease (26.9%). Myocardial infarction (MI) type 4a was revealed in 2.2% of cases. At 4 years, the MACE rates for the whole population and for CHIP and HBR patients were 23.3%, 29.6%, and 27.6%, respectively. CHIP patients had a higher risk of MACEs (29.6% vs. 19.9%, HR 1.69, *p* = 0.032) and cardiac death (11.1% vs. 4.6%, HR 2.50, *p* = 0.048). There were no differences for MI (7.4% vs. 6.6%, *p* = 0.826) and TLR (18.5% vs. 12.6%, *p* = 0.150). HBR patients were also characterized by a higher risk of MACEs (27.6% vs. 21.2%, HR 1.84, *p* = 0.049) and cardiac death (17.1% vs. 1.9%, HR 9.61, *p* < 0.001). There were no differences for MI (7.9% vs. 6.4%, *p* = 0.669) and TLR (11.8% vs. 16.0%, *p* = 0.991). PCI in CHIP and HBR patients is feasible with a low rate of periprocedural complications. Nevertheless, CHIP and HBR patients are at a high risk of future adverse events and require strict surveillance to improve outcomes.

## 1. Introduction

Since 1977, tremendous progress has been made in percutaneous coronary intervention (PCI) techniques and the stents used. Unfortunately, the problem of in-stent restenosis (ISR) is still ongoing despite the invention of drug-eluting stents (DESs) and their continuous improvement. Approximately 10% of all PCI procedures are performed yearly because of ISR [1], and ISR significantly impacts long-term clinical outcomes in patients undergoing PCI. In the NCDR registry having the data of 653,304 patients, PCI due to ISR was performed in 10.2% of cases. Patients undergoing PCI due to ISR were characterized by a worse prognosis. Those patients had a higher rate of major adverse cardiovascular and cerebrovascular events (MACCEs) at 36-month follow up, including a higher incidence of all-cause death, myocardial infarction (MI), or target lesion revascularization (TLR) [2].

In many patients, ISR presents as an acute coronary syndrome without persistent ST-segment elevation. Although DESs have reduced the risk of ISR by 60% compared to bare metal stents (BMSs), the ISR problem remains significant and increases exponentially with the number of reinterventions. Moreover, as the follow up lengthens (i.e., 5–10 years), the late catch-up phenomenon in ISR rates is observed between DESs and BMSs [3].

Various clinical and procedural (considering a lesion, a stent, or a procedure itself) factors are associated with unfavorable outcomes. Comorbidities (diabetes, dyslipidemia, chronic kidney disease) or massive coronary artery calcifications might play a significant role [4,5]. Similarly, the type of stent is also of enormous importance. The thickness of stent struts affects the degree of damage to the arterial wall at the time of implantation and the local blood rheology after stent deployment. As a result, this affects the strength of the inflammatory response at the target lesion, reendothelialization, strut coverage, and neointima formation. Stent underexpansion, malapposition, and the leaving of so-called stent gaps are also linked with an elevated risk of ischemic complications [6,7,8,9]. In 2019, a new restenosis classification was proposed, considering intravascular imaging and available treatment options [10].

With an aging population and improvements in technology, we treat more and more difficult patients both in terms of bleeding as well as ischemic complications. To tackle this issue and better characterize it in further studies, new terms have been defined such as high-bleeding-risk (HBR) patients and patients undergoing complex, high-risk indicated procedures (CHIPs) [11,12]. The abovementioned factors associated with ISR also pose a challenge in CHIP patients to obtain optimal procedural as well as long-term outcomes. In CHIP patients, to obtain the optimal outcome, often, additional interventions are required, such as using rotational atherectomy or orbital atherectomy, which makes the procedures even more difficult and associated with a higher risk of periprocedural complications [13,14]. Moreover, more and more frequently, CHIP patients undergo PCI with the simultaneous use of percutaneous left ventricular assist devices [15,16].

CHIP patients pose the highest challenge in modern PCI; however, in the past 20 years, ischemic events after PCI halved (from 18.4% to 9.1%), and out-of-hospital bleeding doubled (from 2.5% to 5%). The proper identification of HBR patients and bleeding prevention became a priority in modern cardiology. This is because bleeding episodes, even if not linked directly with poor outcomes, evoke worse medication adherence and quality-of-life deterioration [17].

We aimed to characterize the performance and safety of PCI with second-generation sirolimus-eluting stents with a biodegradable polymer in CHIP and HBR patients at a 4-year follow up.

## 2. Materials and Methods

### 2.1. Study Design and Participants

We obtained data retrospectively from the hospital database and analyzed all consecutive patients who underwent PCI with sirolimus-eluting coronary stent Alex Plus (Balton, Poland) implantation between July 2015 and March 2016. We included patients undergoing PCI in the setting of chronic coronary syndrome as well as acute coronary syndrome. Additionally, we differentiated two subgroups, i.e., complex, high-risk indicated procedure (CHIP) patients and high-bleeding-risk (HBR) patients. 

We compared various baseline demographic and clinical characteristics, laboratory data (see Section 2.4), and clinical outcomes (see Section 2.5) at a 4-year follow up in the whole population and in the CHIP and HBR subgroups.

### 2.2. CHIP and HBR Subgroup Criteria

CHIP patients were characterized as having at least one clinical criterion and one anatomical high-risk criterion [11,18]. The clinical criteria were as follows: advanced age (≥75 years), diabetes mellitus, heart failure with left ventricular ejection fraction ≤ 35%, acute coronary syndrome, previous cardiac surgery, peripheral vascular disease, advanced chronic kidney disease (estimated glomerular filtration rate < 30 mL/min/1.73 m^2^), chronic obstructive pulmonary disease, concomitant severe aortic valvulopathy, or severe mitral regurgitation. The anatomical criteria were as follows: unprotected left main disease, degenerated saphenous vein grafts, severely calcified lesions requiring rotational atherectomy, last patent conduit, or chronic total occlusion in a patient with multivessel disease.

HBR patients were characterized based on the Academic Research Consortium for High Bleeding Risk (ARC-HBR). Patients were considered HBR if at least one major or two minor criteria were met [12]. The ARC-HBR criteria were adopted since they provide reliable predictions for major bleeding also in acute coronary syndrome patients [19] and they are not inferior to other scores such as PRECISE-DAPT [20].

### 2.3. Alex Plus Stent Characteristics

The Alex Plus stent platform is made of cobalt–chromium alloy (L605) with a strut thickness of 70 μm. The sirolimus concentration is 1.3 μg/mm^2^, and the drug is released from a biodegradable polymer in a process lasting for 8 weeks [21,22]. The stent’s nominal diameter and length ranges are 2.0–5.0 mm and 8.0–40.0 mm, respectively. The stent can be overexpanded during postdilatation (3.5 mm → 4.3 mm; 4.0 mm → 4.7 mm; 5.0 mm → 6.0 mm).

### 2.4. Data Collection

We gathered demographic, clinical, periprocedural, and laboratory data from the hospital database. We collected information on comorbidities such as arterial hypertension, dyslipidemia, diabetes mellitus, prior PCI, prior MI, chronic kidney disease (defined as eGFR < 60 mL/min/1.73 m^2^), prior coronary artery bypass grafting (CABG), prior stroke, peripheral artery disease, chronic obstructive pulmonary disease (COPD), and smoking. Additionally, we analyzed data on PCI: planned vs. urgent, lesion location, lesion type (A, B1, B2, C according to AHA/ACC classification [23]), and periprocedural complications. Additionally, the SYNTAX (https://syntaxscore.org accessed on 10 April 2023), SYNTAX II [24], and EuroScore II (https://www.euroscore.org accessed on 12 April 2023) were calculated. Moreover, we collected information on echocardiographic parameters (left ventricular ejection fraction (LVEF), left ventricular end-diastolic diameter, intraventricular septal diameter, posterior wall diameter diastolic, left atrial diameter, tricuspid annular plane systolic excursion) and laboratory findings assessed at admission: complete blood count with differential (white blood cells (WBCs), red blood cells (RBCs), hemoglobin (Hgb), platelets (PLTs)), creatine kinase (CK), CK-MB, creatinine, troponin T, estimated glomerular filtration rate (eGFR), glucose, glycated hemoglobin (HbA1c), and lipid profile. We also obtained information on the medications at discharge. We obtained long-term data by phone contact and from the hospital database. If no phone contact was possible, we obtained data on the patient’s status from the National Health Fund.

### 2.5. Study Endpoints

The primary study endpoint was to compare the 4-year rate of major cardiovascular adverse events (MACEs) defined as joined rates of cardiac death, MI, and target lesion revascularization (TLR). The secondary endpoints included all-cause death, cardiac death, MI, and TLR rates at 1, 2, 3, and 4 years.

### 2.6. Statistical Methods

Descriptive statistics are shown as mean, standard deviation, minimum, median, interquartile range, and maximum for continuous variables and as count and percent for categorical variables. The Pearson’s chi-squared test or the Fisher’s exact test was performed to compare categorical variables between two groups (e.g., CHIP and non-CHIP patients). The Fisher’s exact test was used when at least one of the subgroups had count = 0. The Wilcoxon rank sum test was performed to compare continuous variables between two groups (e.g., CHIP and non-CHIP patients). A *p*-value < 0.05 was considered statistically significant.

Kaplan–Meier estimators with 95% CI were calculated to compare the 4-year survival curves for various endpoints between groups (e.g., CHIP and non-CHIP patients). If a given endpoint occurred for a particular patient more than once in a 4-year follow-up period, then survival time was assumed as the time to the first occurrence of this endpoint. Notably, in the case of a MACE (a composite endpoint), the survival time was assumed as the time to the first occurrence of either cardiac death, MI, or TLR.

Univariable and multivariable Cox regression (Cox proportional hazards model) was performed to compare survival rates between groups. The multivariable Cox regression model was chosen in stepwise selection with a backward elimination algorithm with a significance level = 0.1. Results regarding the hazard ratio (HR) and 95% confidence intervals for HR are presented.

Statistical analyses were performed using R software version 4.2.1 (2022-06-23 ucrt)—“Funny-Looking Kid” Copyright (C) 2022, The R Foundation for Statistical Computing Platform: x86_64-w64-mingw32/x64 (64-bit).

## 3. Results

### 3.1. Patient Inclusion

In the analyzed period, we identified 872 PCI procedures. Amongst these, we identified 250 patients with 307 Alex Plus stent labels in the procedure books. However, in four patients (five stents), Alex Plus stents were not implanted (one device failure—no possibility to deliver the stent to the target lesion due to calcification and tortuosity, four stents not implanted due to fatal cardiac arrest). Moreover, we excluded 14 patients (20 stents) due to in-hospital death unrelated to the sirolimus-eluting stent deployment. Ultimately, we analyzed 232 patients in whom 282 stents were implanted. Additionally, we identified 81 patients meeting the CHIP criteria and 76 patients meeting the HBR criteria (Figure 1).

### 3.2. Baseline Characteristics

In the whole population, the mean age was 68 ± 11 years and 23.7% were females. The PCI procedures in 38.4% of cases were performed in the acute setting. The following comorbidities were the most prevalent: arterial hypertension (91.8%), dyslipidemia (76.3%), prior PCI (56%), prior MI (48.7%), and diabetes type 2 (41.8%). The mean LVEF was 49.5 ± 10.5% (Table 1). 

As stated earlier, the CHIP definition was met by 81 patients. The median number of fulfilled clinical criteria was four (IQR 2–7), and the median number of fulfilled anatomical criteria was two (IQR 1–3). The CHIP patients were older (70 ± 11 vs. 67 ± 11 years, *p* = 0.027) and characterized by higher rates of MI as a reason for PCI (*p* = 0.003), diabetes (*p* = 0.002), and dyslipidemia (*p* = 0.008). Similarly, the HBR patients were older (77 ± 8 vs. 63 ± 9 years, *p* < 0.001) and characterized by higher rates of diabetes type 2 (*p* < 0.001) and chronic kidney disease (*p* < 0.001). Differences between the CHIP vs. non-CHIP and the HBR and non-HBR subgroups are presented in Appendix A. 

The laboratory findings are shown in Table 2. Additional data are also presented in Appendix A. The CHIP patients were characterized by lower red blood cells (*p* = 0.038) and lower LDL values (*p* = 0.042). At the same time, the HBR patients were characterized by lower red blood cells (*p* < 0.001), hemoglobin values (*p* < 0.001), total cholesterol (*p* < 0.001), LDL values (*p* < 0.001), triglycerides (*p* = 0.007) and eGFR (*p* < 0.001).

### 3.3. Procedure Characteristics

In the whole study population, most lesions were located in the right coronary artery (38.8%), followed by the left anterior descending artery (31%) and the left circumflex artery (26.3%). The lesions were mostly complex—type C lesions accounted for 39.2% of cases. Coronary bifurcations were treated in 9.9% of cases. The mean SYNTAX PCI score was 32.9 ± 11.0 (Table 3). Most procedures were performed from radial access (83.2%) using a 6F guiding catheter (95.7%). The lesions were mostly predilated (61.6%), and postdilatation was performed in 37.9% of cases. The mean nominal parameters of the Alex Plus stent were 3.2 ± 0.5 mm × 21.2 ± 10.9 mm. The device success was 99.6% (in one case, a second stent was required due to heavy calcifications). Additional stents were deployed in 39% of cases due to edge dissection (6.9%), side branch stenting (5.2%), or diffuse disease (26.9%). MI type 4a was revealed in 2.2% of cases (Table 3). 

The CHIP patients had more procedures within the left main (0.7 vs. 9.9%), more complex lesions (type C: 31.8% vs. 53.1%, *p* = 0.014), and larger implanted Alex Plus stents (3.1 ± 0.5 × 18.3 ± 7.0 mm vs. 3.3 ± 0.5 × 26.7 ± 14.3 mm) as well as more frequent additional stent implantations (12.7% vs. 87.7%, *p* < 0.001). The rate of coronary dissection was higher in the CHIP patients than in the non-CHIP ones (13.6% vs. 3.3%, *p* = 003), with similar rates of MI type 4a (1.2% vs. 2.7%, *p* = 0.66). There were no statically significant differences in periprocedural complications in the HBR vs. the non-HBR groups (dissection: 3.9% vs. 8.3%, *p* = 0.216 and MI type 4a: 1.3 vs. 2.6%, *p* = 0.999) (Appendix A).

The medications at discharge are presented in Table 4 as well as Appendix A. All patients received acetylsalicylic acid and P2Y12 inhibitors (clopidogrel—92.2%). The other key medications were as follows: beta-blockers (96.1%), ACEI/ARB (97.4%), and statins (99.1%).

### 3.4. Four-Year Outcomes

The detailed rates of MACEs, death, cardiac death, MI, and TLR at 12, 24, 36, and 48 months are presented in Table 5 and Figure 2. At 4 years, the MACE rates for the whole population, CHIP, HBR, and CHIP + HBR were 23.3%, 29.6%, 27.6%, and 31.4%, respectively. 

CHIP patients had a higher risk of MACEs (29.6% vs. 19.9%, HR 1.69, *p* = 0.032) and cardiac death (11.1% vs. 4.6%, HR 2.50, *p* = 0.048). There were no differences for MI (7.4% vs. 6.6%, *p* = 0.826) and TLR (18.5% vs. 12.6%, *p* = 0.150). At 12 months, in non-CHIP patients (*n* = 151), the rates were as follows: cardiac death—3.3% (*n* = 5), TLR—5.3% (*n* = 8), and MI—3.3% (*n* = 5). Similarly, at 48 months, in non-CHIP patients (*n* = 151), the rates were as follows: cardiac death—7 (4.6%), TLR—12.5% (*n* = 19), and MI—6.6% (*n* = 10).

HBR patients were also characterized by a higher risk of MACEs (27.6% vs. 21.2%, HR 1.84, *p* = 0.049) and cardiac death (17.1% vs. 1.9%, HR 9.61, *p* < 0.001). There were no differences for MI (7.9% vs. 6.4%, *p* = 0.669) and TLR (11.8% vs. 16.0%, *p* = 0.991).

### 3.5. Multivariable Cox Regression Analysis

Finally, the Cox analysis identified predictive factors for MACEs and TLR at 48 months. The final results of the multivariable analyses are provided in Table 6 for MACEs, and those for TLR are presented in Table 7 (univariable analyses are presented in Appendix A). For the whole population, the predictive factors of MACEs were lesion in the LM (HR 3.88), calcification (HR 2.70), second stent implantation (HR 2.06), EuroScore II > 5% (HR 2.87), and prior PCI (HR 2.09). 

In CHIP patients, prior CABG (HR 3.02) and chronic kidney disease (HR 5.07) increased the MACE rate, but beta-blockers significantly reduced the risk (HR 0.001). In the case of HBR patients, EuroScore II 3–5% (HR 4.12), cardiogenic shock (HR 13.0), and hypoglycemic drug use (HR 5.30) were independent predictors for MACEs.

## 4. Discussion

Our study showed that PCI with Alex Plus stent in CHIP and HBR patients is feasible with a low rate of periprocedural complications and a device success of over 99%. Nevertheless, CHIP and HBR patients are at high risk of future adverse events and require strict surveillance to improve outcomes. 

In our study, at 4 years, the MACE rates for the whole population, CHIP, HBR, and CHIP + HBR were 23.3%, 29.6%, 27.6%, and 31.4%, respectively. When analyzing the results in detail, we can also distinguish the non-complex group of patients treated with Alex Plus (non-CHIP). At 12 months, in non-CHIP patients (*n* = 151), the rates were as follows: cardiac death—3.3% (*n* = 5), TLR—5.3% (*n* = 8), and MI—3.3% (*n* = 5). Similarly, at 48 months, in non-CHIP patients (*n* = 151), the rates were as follows: cardiac death—7 (4.6%), TLR—12.5% (*n* = 19), and MI—6.6% (*n* = 10). Nevertheless, this non-CHIP subgroup included HBR patients, which could also negatively influence the results. One of the reasons could be the limited use of new potent antiplatelets in HBR patients [25]. The first year’s results are important since, as Eccleston et al. showed, unplanned early hospitalization following PCI, particularly in <30 days, was linked with a significantly higher incidence of MACEs at long-term follow up [26].

Sirolimus-eluting stents (SESs) have already been proven to be effective. In STEMI patients, Sakurai et al. confirmed that primary PCI with SESs was linked with reduced all-cause death and TLR rates without increased rates of recurrent MI or definite stent thrombosis compared to BMSs. Up to 48 months, SESs secured a significant reduction in TLR (OR 0.44, *p* < 0.001) and all-cause death (OR 0.62, *p* = 0.049) compared to BMSs [27]. Nevertheless, nowadays, BMSs are rarely used, and we should compare Alex Plus’s performance with that of other second-generation SESs. Table 8 summarizes studies showing the long-term results of patients undergoing PCI with SESs identified in PubMed in the last 10 years.

As stressed earlier in our study, the CHIP group was characterized by the worst outcomes, with a cardiac death rate of 11.1% and a TLR rate of 18.5% at 4 years. This is in accordance with the available literature since CHIP patients are characterized by both worse periprocedural as well as long-term outcomes [28]. Nevertheless, in our paper, the CHIP patients were not characterized by high periprocedural complications. Regarding periprocedural complications, we should not forget about bleeding, especially in HBR patients. In our study, 83.2% of procedures were performed from radial access. Other authors show that one of the options to decrease the bleeding risk even further is to use distal radial artery access. Such access might be beneficial for HBR patients but is also feasible in CHIP patients [29,30].

**Table 8 jcm-12-05313-t008:** Clinical outcomes at long-term follow in studies on PCI with SESs.

Study	Comments	Death	Cardiac Death	TLR	MI
Our study: whole		10.8%	6.9%	14.7%	6.9%
Our study: Non-complex	Including HBR	8.6%	4.6%	12.5%	6.6%
Our study: CHIP		14.8%	11.1%	18.5%	7.4%
Murray [31]	-	18.8%	-	-	-
Riku [32]	5 yrs FU non-complex vs. complex PCI	7.0 vs. 12.2%	2.9 vs. 5.4%	9.8 vs. 22%	1.4 vs. 3.1%
Buiten [33]	3 yrs FU SES/EES/ZES	-	2.4%/2.5%/2.5%	7%/9.5%/10%	3.3%/3.9%/4.2%
Maeng [34]	EES/SES	12/9.5%	8.3/4.8%	5.6/9.5%	1.9/7.6%
Olesen [35]	5 yrs FU ZES/SES	16/17.9%	6.5/7.1%	14.8/4.8%	8.9/7.1%
Wijns [36]	ZES/SES	5.5/6%	2.9/3.4%	9/8.6%	4.6/5.8
Sato [37]	5 yrs FU DM/non-DM	7.1/8.9%	2.4/2.7%	9.4/8.9%	4.7/0%
Stefanini [38]	BP-DES/DP-SES	9.3/10%	5.2/5.9%	12/13.7%	6/6.8%

TLR—target lesion revascularization; MI—myocardial infarction; FU—follow up; PCI—percutaneous coronary intervention; SES—sirolimus-eluting stent; ZES—zotarolimus-eluting stent; EES—everolimus-eluting stent; DM—diabetes mellitus; BP—biodegradable polymer; DP—durable polymer.

As presented in the identified studies with follow up from 3 to 5 years (follow up precisely 4 years), the cardiac death rate of patients with SESs deployed was 2.9–7.1% (3.4–5.9%); the MI rate was 1.4–7.6% (5.8–7.6%), and the TLR rate was 7–22% (8.6–13.7%). The results for Alex Plus in non-complex PCI (as most studies represented similar populations) fall within those ranges, i.e., cardiac death rate—4.6%, MI rate—6.6%, and TLR rate—12.5%. In our study, the highest TLR rate was observed in the CHIP group (18.5%), and as presented in the study by Riku et al., the TLR rate in the complex PCI group was 22% [32].

Interestingly, Riku et al. also presented outcomes up to 10 years [32]. As one could presume, the TLR rate was significantly higher in the complex PCI group than in the non-complex PCI group (29.4% vs. 13.0%, *p* = 0.001). Late TLR cases were observed over 10 years at a rate of 2.4% per year in the complex PCI group and at 1.1% per year in the non-complex PCI group. The cardiac death rate was higher in the complex PCI group than in the non-complex PCI group, particularly after 4 years (15.8% vs. 7.5%, *p* = 0.031). Worth stressing is the fact that sudden death was the major reason for cardiac death beyond 4 years in patients from the complex PCI group. 

Brenner et al. proposed a more straightforward CHIP-PCI classification, dividing patients into low-risk, intermediate-risk, and high-risk PCI. The 12-month all-cause death rates in those three groups were 1.24%, 2.47%, and 10.86%, respectively (*p* < 0.001) [39].

The Alex Plus is a sirolimus-eluting stent with a biodegradable polymer. This may be an advantage since, recently, Mattke et al. showed that biodegradable polymer (BP) SESs compared to durable polymer (DP) EESs, were associated with 2603 fewer deaths per one million patients over 48 months. This corresponded with a relative risk reduction of 6% [40]. Although de Waha et al. showed that BP DESs and DP SESs were characterized by comparable clinical outcomes at 4 years, the stent thrombosis rate was significantly lower in patients with BP DESs [41].

Finally, it is crucial to identify risk factors that could negatively impact the outcomes in patients undergoing PCI with second-generation DESs. Here, we identified the following. For the whole population, the predictive factors of MACEs were lesion in the left main (HR 3.88), calcifications (HR 2.70), second stent implantation (HR 2.06), EuroScore II > 5% (HR 2.87), and prior PCI (HR 2.09). In CHIP patients, prior CABG (HR 3.02) and chronic kidney disease (HR 5.07) increased the MACE rate, but beta-blockers significantly reduced the risk (HR 0.001). In the case of HBR patients, EuroScore II 3–5% (HR 4.12), cardiogenic shock (HR 13.0), and hypoglycemic drug use (HR 5.30) were independent predictors increasing the MACE risk. The predictive factors for TLR were a bit different. In the whole population, there were lesions in the left main (HR 14.9), calcifications (HR 3.07), and second stent implantation (HR 4.09). In the CHIP subgroup, prior CABG (HR 3.94) was the only predictive factor, and in the HBR subgroup, there were postdilatation (HR 5.62), smoking (HR 5.70), and use of alpha-adrenolytics (HR 5.22). In the HBR group, male sex (HR 0.15) was the only factor decreasing the TLR risk.

Fujimoto et al. showed that pulmonary disease, active malignancy, unstable hemodynamics, hemodialysis, left ventricular ejection fraction, and valvular disease significantly increased the MACE risk [42]. In their other paper, the same group showed that frailty (OR 2.04, 95% CI 1.10–3.75, *p* = 0.022), unstable hemodynamics (OR 5.75, 95% CI 1.21–27.20, *p* = 0.027), and immunosuppressive drugs (OR 3.04, 95% CI 1.25–7.38, *p* = 0.014) were linked with the increased risk of major complications in CHIP patients [43]. Interestingly, Guldener et al., applying machine learning and self-organizing maps, identified additional ISR risk factors such as age, BMI, dyslipidemia, chronic artery occlusion, clinical presentation (STEMI vs. NSTEMI vs. elected PCI), baseline TIMI flow in the treated coronary artery, prior PCI, lesion length, and residual diameter stenosis post-PCI [44].

We think that future studies, especially performed on a large scale, should answer the question of whether all drugs (sirolimus, everolimus, zotarolimus, biolimus A9) are equally effective when used during PCI in CHIP and HBR patients.

### Study Limitations

Our study has the recognized limitations of registries and observational studies. The lack of randomization might have led to selection bias, even if partially mitigated by the consecutive patient enrolment. Moreover, the moderate size of this study’s population and the limitations in follow-up data gathering could have also influenced the results.

## 5. Conclusions

CHIP patients pose the highest challenge in modern PCIs. However, in the past 20 years, ischemic events after PCI halved (from 18.4% to 9.1%), and out-of-hospital bleeding doubled (from 2.5% to 5%). Our study showed that PCI with a second-generation sirolimus-eluting stent in CHIP and HBR patients is feasible with a reasonable rate of periprocedural complications, including low rates of MI type 4a. The MACE rates at 4 years support the performance and safety of this stent in those populations. Nevertheless, CHIP and HBR patients are at high risk of future adverse events and require strict surveillance to improve outcomes.

## Figures and Tables

**Figure 1 jcm-12-05313-f001:**
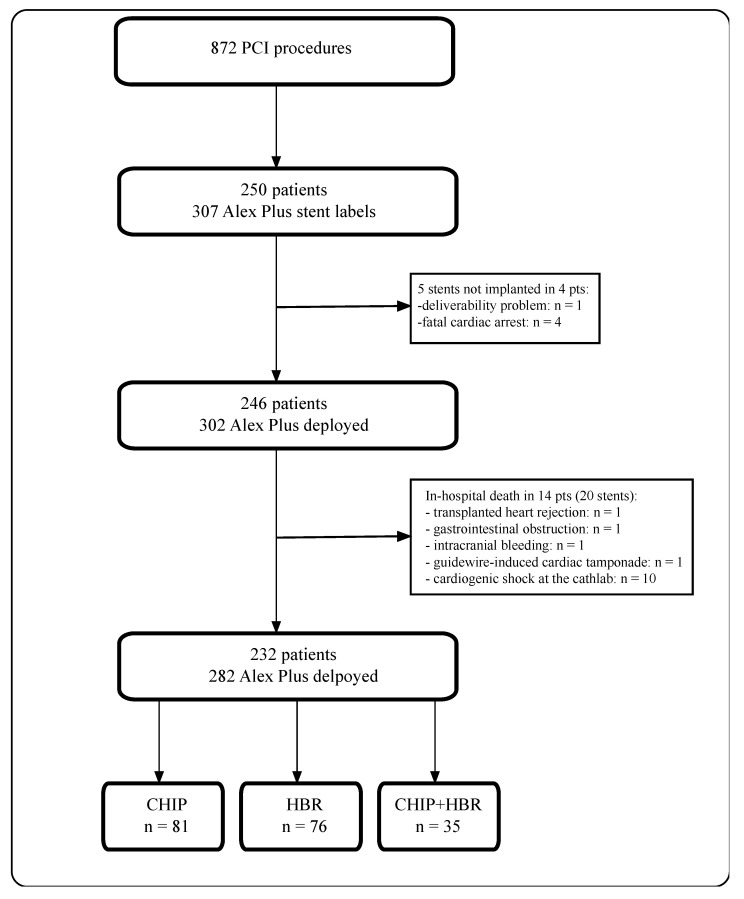
Study flow chart.

**Figure 2 jcm-12-05313-f002:**
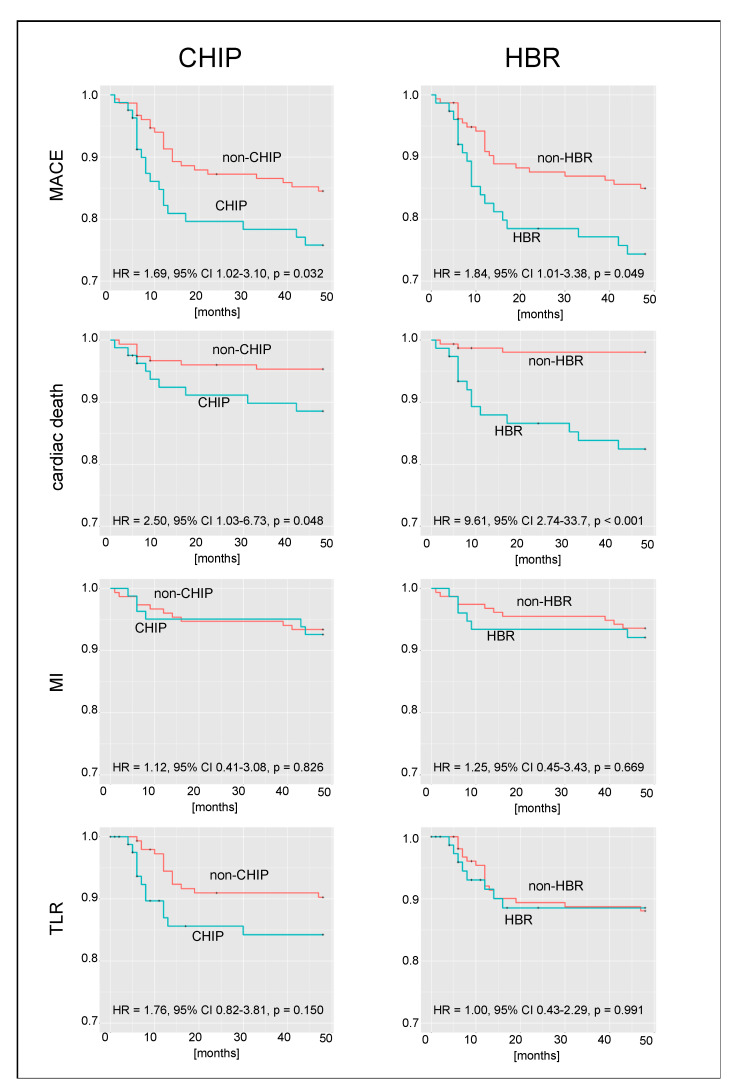
Kaplan–Meier curves showing event-free survival in CHIP and HBR patients. MI—myocardial infarction; TLR—target lesion revascularization.

**Table 1 jcm-12-05313-t001:** Baseline characteristics.

Parameter	Whole Population *n* = 232 (%)	CHIP *n* = 81 (%)	HBR *n* = 76 (%)
Females	55 (23.7)	22 (27)	25 (33)
Age [years]	68 ± 11	70 ± 11	77 ± 8
Acute coronary syndrome type at presentation			
UA	31 (13.4)	13 (16.0)	11 (14.5)
NSTEMI	26 (11.2)	16 (19.8)	11 (14.5)
STEMI	32 (13.8)	13 (16.0)	10 (13.2)
Cardiogenic shock	6 (2.6)	3 (3.7)	3 (3.9)
Arterial hypertension	213 (91.8)	76 (93.8)	71 (93.4)
Diabetes type 2	97 (41.8)	45 (55.6)	44 (57.9)
Dyslipidemia	177 (76.3)	70 (86.4)	61 (80.3)
Prior myocardial infarction	113 (48.7)	42 (51.9)	43 (56.6)
Prior PCI	130 (56.0)	48 (59.3)	49 (64.5)
Prior CABG	22 (9.5)	12 (14.8)	11 (14.5)
Chronic kidney disease	42 (18.1)	16 (19.8)	27 (35.5)
Prior stroke	17 (7.3)	6 (7.4)	10 (13.2)
Peripheral artery disease	25 (10.8)	12 (14.8)	9 (11.8)
Chronic obstructive pulmonary disease	13 (5.6)	6 (7.4)	6 (7.9)
Echocardiographic parameters			
LVEDd [mm]	50.4 ± 9.0	52.0 ± 7.9	51.5 ± 8.5
IVSd [mm]	11.4 ± 2.1	11.4 ± 2.4	11.6 ± 1.7
PWDd [mm]	10.5 ± 1.6	10.6 ± 1.9	10.5 ± 1.6
LA [mm]	40.4 ± 5.9	41.0 ± 6.0	43.2 ± 5.8
TAPSE [mm]	22.0 ± 4.3	22.2 ± 4.5	21.3 ± 4.8
LVEF [%]	49.5 ± 10.5	48.9 ± 10.4	47.0 ± 12.2
Severe mitral regurgitation	6 (3.1)	2 (2.7)	4 (6.1)
Severe aortic regurgitation	1 (0.5)	0	0
Severe aortic stenosis	4 (2.1)	3 (4.1)	3 (4.5)

UA—unstable angina; NSTEMI—non-ST-elevation myocardial infarction; STEMI—ST-elevation myocardial infarction; PCI—percutaneous coronary intervention; CABG—coronary artery bypass grafting; LVEDd—left ventricular end-diastolic diameter; IVSd—intraventricular septal diameter; PWDd—posterior wall diastolic diameter; LA—left atrium; TAPSE—tricuspid annular plane systolic excursion; LVEF—left ventricular ejection fraction.

**Table 2 jcm-12-05313-t002:** Laboratory test findings.

Parameter	Whole Population *n* = 232	CHIP *n* = 81	HBR *n* = 76
White blood cells [10^9^/L]	8.5 ± 2.7	8.6 ± 2.3	8.0 ± 2.2
Hemoglobin [g/dL]	13.4 ± 1.7	13.1 ± 1.7	12.1 ± 1.9
Red blood cells [10^12^/L]	4.4 ± 0.5	4.3 ± 0.5	4.1 ± 0.6
Platelets [10^9^/L]	222.9 ± 65	215.8 ± 63	222.5 ± 64.7
Glucose [md/dL]	136.4 ± 64.9	154.8 ± 75.3	148.3 ± 77.4
HbA1c [%]	6.3 (6.0–7.3)	6.6 (6.1–7.3)	6.4 (6.0–7.1)
Total cholesterol [md/dL]	163.9 ± 50.9	161.5 ± 60.6	143.4 ± 41.4
HDL [md/dL]	45.7 ± 14.6	44.0 ± 43.5	44.7 ± 13.1
LDL [md/dL]	89.8 ± 40.5	83.0 ± 42.6	76.0 ± 35.8
Triglycerides [md/dL]	142 ± 33.9	170.7 ± 89.9	113.1 ± 62.3
Creatine [md/dL]	1.1 ± 0.7	1.2 ± 0.9	1.4 ± 0.8
eGFR	70.5 ± 23.2	67.4 ± 23.4	56.6 ± 20.4
TnI at admission [ng/mL]	108 (15.8–235)	211.5 (28.2–754.5)	62.5 (19.0–1446)
Max. TnI [ng/mL]	1110 (49.8–11,573)	1263 (88.2–13,371)	372 (44.0–8802)
CK	134.5 (84–326)	169 (75–319)	118 (72–334)
CK max	173 (90–473)	183 (80–390)	156 (74–363)
CK-MB	18 (13.5–30)	20 (13–34.5)	18 (13–29.8)
CK-MB max	22.5 (15–48.5)	26.5 (14.2–68)	23.5 (14.2–40)

Results presented as mean ± standard deviation or median (interquartile range); CK—creatine kinase.

**Table 3 jcm-12-05313-t003:** Lesion and procedure characteristics.

Parameter	Whole Population *n* = 232 (%)	CHIP *n* = 81 (%)	HBR *n* = 76 (%)
Lesion location			
LM	9 (3.9)	8 (9.9)	5 (6.6)
LAD	72 (31)	21 (25.9)	31 (40.8)
LCx	61 (26.3)	16 (19.8)	14 (8.4)
RCA	90 (38.8)	36 (44.4)	29 (38.2)
VG	6 (2.6)	6 (7.4)	2 (2.6)
Lesion type			
A	38 (16.4)	12 (14.8)	14 (18.4)
B1	66 (28.4)	17 (21)	24 (31.6)
B2	37 (15.9)	9 (11.1)	8 (10.5)
C	91 (39.2)	43 (53.1)	30 (39.5)
Heavy calcification	18 (7.8)	9 (11.1)	6 (7.9)
Coronary bifurcation	23 (9.9)	12 (14.8)	6 (7.9)
SYNTAX	13.9 ± 8.7	16.0 ± 8.4	14.8 ± 9.2
SYNTAX II PCI	32.9 ± 11.0	35.6 ± 10.1	40.3 ± 10.5
SYNTAX II CABG	29.1 ± 10.8	29.9 ± 10.5	28.4 ± 10.5
EuroScore II	1.6 (0.9–3.3)	2.5 (1.3–4.3)	3.6 (1.8–6.6)
Lesion predilatation	143 (61.6)	47 (58.0)	48 (63.2)
Stent diameter [mm]	3.2 ± 0.5	3.3 ± 0.5	3.1 ± 0.5
Stent length [mm]	21.2 ± 10.9	26.7 ± 14.3	20.3 ± 8.8
Stent pressure [atm]	15.3 ± 2.7	15.5 ± 2.6	15.4 ± 2.5
2nd stent	90 (39)	71 (87.7)	34 (44.7)
Stent postdilatation	88 (37.9)	34 (42.0)	54 (35.8)
Access site			
Transradial	193 (83.2)	64 (79)	56 (73.7)
Transfemoral	43 (18.5)	19 (23.5)	20 (26.3)
Guiding catheter			
6F	222 (95.7)	76 (93.8)	71 (93.4)
7F	11 (4.7)	5 (6.2)	5 (6.6)
Dissection	16 (6.9)	11 (13.6)	3 (3.9)
MI type 4a	5 (2.2)	1 (1.2)	1 (1.3)

LM—left main; LAD—left anterior descending artery; LCx—left circumflex artery; RCA—right coronary artery; VG—vein graft.

**Table 4 jcm-12-05313-t004:** Medications at discharge.

Parameter	Whole Population *n* = 232 (%)	CHIP *n* = 81 (%)	HBR *n* = 76 (%)
Acetylsalicylic acid	232 (100)	81 (100)	76 (100)
P2Y12			
Clopidogrel	214 (92.2)	74 (91.4)	72 (94.7)
Prasugrel	1 (0.4)	1 (1.2)	0
Ticagrelor	17 (7.3)	6 (7.4)	4 (5.3)
Beta-blocker	223 (96.1)	80 (98.8)	73 (96.1)
Ca-blocker	53 (22.8)	21 (25.9)	12 (15.8)
Angiotensin-converting enzyme inhibitor	190 (81.9)	66 (81.5)	60 (78.9)
Angiotensin receptor blocker	36 (15.5)	14 (17.3)	13 (17.1)
Diuretic	125 (53.9)	51 (63)	61 (80.3)
Mineralocorticoid receptor antagonist	48 (20.7)	16 (19.8)	20 (26.3)
Nitrates	13 (5.6)	8 (9.9)	7 (9.2)
Vitamin K antagonist	17 (7.3)	4 (4.9)	16 (21.1)
Novel oral anticoagulant	11 (4.7)	5 (6.1)	10 (13.1)
Statin	230 (99.1)	81 (100)	76 (100)
Hypoglycemic medications	62 (26.7)	29 (35.8)	23 (30.3)
Insulin	33 (14.2)	20 (24.7)	18 (23.7)

**Table 5 jcm-12-05313-t005:** Outcomes at 4 years.

Year	Death	Cardiac Death	TLR	MI	MACE
Whole population (*n* = 232)					
1st year	17 (7.3)	11 (4.7)	18 (7.8)	9 (3.9)	27 (11.6)
2nd year	19 (8.2)	13 (5.6)	28 (12.1)	11 (4.7)	39 (16.8)
3rd year	21 (9.1)	15 (6.5)	31 (13.4)	11 (4.7)	44 (18.9)
4th year	25 (10.8)	16 (6.9)	34 (14.7)	16 (6.9)	54 (23.3)
CHIP (*n* = 81)					
1st year	9 (11.1)	6 (7.4)	10 (12.3)	4 (4.9)	14 (17.3)
2nd year	10 (12.3)	7 (8.6)	13 (16.1)	4 (4.9)	18 (22.2)
3rd year	11 (13.6)	8 (9.9)	14 (17.3)	4 (4.9)	20 (24.7)
4th year	12 (14.8)	9 (11.1)	15 (18.5)	6 (7.4)	24 (29.6)
HBR (*n* = 76)					
1st year	11 (14.5)	9 (11.8)	6 (7.9)	5 (6.6)	13 (17.1)
2nd year	13 (17.1)	10 (13.2)	8 (10.5)	5 (6.6)	16 (21.1)
3rd year	15 (19.7)	12 (15.8)	9 (11.8)	5 (6.6)	19 (25.0)
4th year	17 (22.4)	13 (17.1)	9 (11.8)	6 (7.9)	21 (27.6)
CHIP + HBR (*n* = 35)					
1st year	8 (22.9)	6 (17.1)	4 (11.4)	3 (8.6)	8 (22.9)
2nd year	10 (28.6)	7 (20.0)	4 (11.4)	3 (8.6)	9 (25.7)
3rd year	12 (34.3)	8 (22.9)	4 (11.4)	3 (8.6)	9 (25.7)
4th year	14 (40.0)	9 (25.7)	4 (11.4)	4 (11.4)	11 (31.4)

Values presented as *n* (%). TLR—target lesion revascularization; MI—myocardial infarction; MACE—major adverse cardiovascular event; CHIP—complex, high-risk indicated procedure; HBR—high bleeding risk.

**Table 6 jcm-12-05313-t006:** Multivariable analysis for MACEs.

Characteristic	Multivariable Analysis for MACEs		
	HR	95% CI	*p*
Whole population (*n* = 232)			
Lesion in left main	3.88	1.42, 10.6	0.008
Calcification	2.70	1.17, 6.23	0.020
Second stent	2.06	1.11, 3.84	0.023
EuroScore II			
<3	—	—	
3–5	1.92	0.83, 4.42	0.125
>5	2.87	1.32, 6.23	0.008
Prior PCI	2.09	1.03, 4.21	0.040
CHIP (*n* = 81)			
Prior CABG	3.02	1.02, 8.92	0.045
Chronic kidney disease	5.07	1.44, 17.9	0.011
Beta-blocker	0.00	0.00, 0.06	<0.001
Diuretics	3.02	0.80, 11.4	0.103
HBR (*n* = 76)			
EuroScore II			
<3	—	—	
3–5	4.12	1.16, 14.6	0.028
>5	2.19	0.67, 7.19	0.196
Cardiogenic shock	13.0	1.99, 85.4	0.007
Smoking	2.62	0.94, 7.26	0.065
Hypoglycemic drugs	5.30	1.90, 14.8	0.001

CHIP—complex, high-risk indicated procedure; HBR—high bleeding risk; PCI—percutaneous coronary intervention; CABG—coronary artery bypass grafting.

**Table 7 jcm-12-05313-t007:** Multivariable analysis for TLR.

Characteristic	Multivariable Analysis for TLR		
	HR	95% CI	*p*
Whole population (*n* = 232)			
Lesion in left main	14.9	3.95, 56.2	<0.001
Calcification	3.07	1.12, 8.37	0.029
Second stent	4.09	1.72, 9.75	0.001
CHIP (*n* = 81)			
Prior CABG	3.94	1.15, 13.5	0.029
HBR (*n* = 76)			
Male sex	0.15	0.03, 0.71	0.017
Postdilatation	5.62	1.06, 29.9	0.043
Smoking	5.70	1.02, 31.7	0.047
Alpha-adrenolytic	5.22	1.04, 26.3	0.045

CHIP—complex, high-risk indicated procedure; HBR—high bleeding risk; CABG—coronary artery bypass grafting.

## Data Availability

Data are available from the corresponding author on request.

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
