# Peer review of "Clinical Outcomes and Prognostic Factors in Complex, High-Risk Indicated Procedure (CHIP) and High-Bleeding-Risk (HBR) Patients Undergoing Percutaneous Coronary Intervention with Sirolimus-Eluting Stent Implantation: 4-Year Results"

_jcm, 2023, doi:10.3390/jcm12165313_

Round 1

Reviewer 1 Report

Main Evaluations and Issues:

1. Abstract: While the abstract provides a brief overview of the study, it does not adequately summarize the main findings of the paper. We suggest that you revise the abstract to provide a clearer summary of your results.

2. Introduction: The Introduction section does not provide sufficient background information or context to help readers understand the significance of your research. We suggest that you consider adding more detail to help readers better understand the importance of your work.

3. Methodology: The methodology is not presented in sufficient detail, and the organization could be improved to enhance the readability and clarity of the paper.

4. Supporting Literature: We did not find sufficient supporting literature for your research topic, particularly in the sections discussing the theoretical framework and the methods used. Please consider incorporating additional relevant literature and improving the citations.

5. Results: While you provided some analysis of the results, the conclusions do not appear to be particularly insightful or thought-provoking. We suggest that you review your analysis and generate more insightful claims and implications from your findings.

6. Future Research: The manuscript does not provide any direction on future research or the next steps following this study's completion. Please consider providing some guidance in this area.

7. Importance of Conclusions: Although the conclusions drawn are adequately supported by the data, we believe they could be presented in a more appealing and engaging manner to better convey their significance and impact.

8. References: There were some minor issues with the references, including incomplete citations or outdated sources. Please check the references carefully and make the necessary corrections.

9. Grammar: It is noted that your manuscript needs careful editing by someone with expertise in technical English editing paying particular attention to English grammar, spelling, and sentence structure so that the goals and results of the study are clear to the reader.

Author Response

  1. Abstract: While the abstract provides a brief overview of the study, it does not adequately summarize the main findings of the paper. We suggest that you revise the abstract to provide a clearer summary of your results.

Answer: The abstract word limitation is 200 words, so we are not able to include more data. Nevertheless, we added exact rates of endpoints at 4 years such as MACE, MI, TLR, and not only using HR values. “CHIP patients had a higher risk of MACE (29.6% vs. 19.9%, HR 1.69, p = 0.032) and cardiac death (11.1% vs. 4.6%, HR 2.50, p = 0.048). There were no differences for MI (7.4% vs. 6.6%, p = 0.826) and TLR (18.5% vs. 12.6%, p = 0.150). HBR patients also characterized a higher risk of MACE (27.6% vs. 21.2%, HR 1.84, p = 0.049) and cardiac death (17.1% vs. 1.9%, HR 9.61, p < 0.001). There were no differences for MI (7.9% vs. 6.4%, p = 0.669) and TLR (11.8% vs. 16.0%, p = 0.991).”

  1. Introduction: The Introduction section does not provide sufficient background information or context to help readers understand the significance of your research. We suggest that you consider adding more detail to help readers better understand the importance of your work.

Answer: We included more elaboration on CHIP and HBR patients adding: “Also, abovementioned factors associated with ISR pose a challenge in CHIP patients to obtain optimal procedural as well as long term outcomes. In CHIP patient to obtain the optimal outcome often additional interventions are required such as using rotational atherectomy or orbital atherectomy what makes the procedures even more difficult and associated with higher risk of periprocedural complications [13, 14]. Moreover, more and more frequently CHIP patients undergo PCI with simultaneous use of percutaneous left ventricle assist devices [15, 16]. As mentioned above, CHIP patients pose the highest challenge in modern PCI, especially considering the applied technique. However, in the past 20 years, ischemic events after PCI halved (from 18.4% to 9.1%), and out-of-hospital bleeding doubled (from 2.5% to 5%). Proper identification of HBR patients and bleeding prevention became a priority in modern cardiology. This is because bleeding episodes, even if not linked directly with poor outcomes, evoke worse medication adherence and quality of life deterioration [17]”

  1. Methodology: The methodology is not presented in sufficient detail, and the organization could be improved to enhance the readability and clarity of the paper.

Answer: For clarity we made a new subsection (2.2) with definition/criteria of CHIP and HBR. We also added new references to support the chosen criteria. We think that the other parts are rather clear. Section 2.1 shortly presents the study design, ethical issues are at the end of the paper (according to MDPI house style), section 2.3 provides Alex Plus characteristics with references to the literature, section 2.4 in depth provide info on collected data, including each parameter (which comorbidities, which periprocedural aspects, which lab tests, which echo parameters were included). Here, we added info on long-term data gathering. Section 2.5 clearly defines study endpoints. Section 2.6 shows statistical methods in depth.

  1. Supporting Literature: We did not find sufficient supporting literature for your research topic, particularly in the sections discussing the theoretical framework and the methods used. Please consider incorporating additional relevant literature and improving the citations.

Answer: We added new references to support the chosen criteria for CHIP and HBR.

  1. Results: While you provided some analysis of the results, the conclusions do not appear to be particularly insightful or thought-provoking. We suggest that you review your analysis and generate more insightful claims and implications from your findings.

Answer: Our aim was to show performance and safety of Alex Plus stent at 4 years, also, in CHIP and HBR population. We made some changes improving clarity and easiness of reading; however, we would not like to modify this section in other way.

  1. Future Research: The manuscript does not provide any direction on future research or the next steps following this study's completion. Please consider providing some guidance in this area.

Answer: We added the following “We think that the future studies, especially performed on a large scale, should answer the question if all drugs (sirolimus, everolimus, zotarolimus, biolimus A9) are equally effective when using during PCI in CHIP and HBR patients.”

  1. Importance of Conclusions: Although the conclusions drawn are adequately supported by the data, we believe they could be presented in a more appealing and engaging manner to better convey their significance and impact.

Answer: We modified slightly the conclusions.” CHIP patients pose the highest challenge in modern PCI. However, in the past 20 years, ischemic events after PCI halved (from 18.4% to 9.1%), and out-of-hospital bleeding doubled (from 2.5% to 5%). Our study showed that PCI with 2nd generation sirolimus eluting stent in CHIP and HBR patients is feasible with a reasonable rate of periprocedural complications, including low rates of MI type 4a. MACE rates at 4 years support the performance and safety of this stent in those populations. Nevertheless, CHIP and HBR patients are at high risk of future events and require strict surveillance to improve out-comes”

  1. References: There were some minor issues with the references, including incomplete citations or outdated sources. Please check the references carefully and make the necessary corrections.

Answer: We checked and updated the references (in some cases incomplete citations result from papers being ahead of print). We added also some more references being associated with CHIP and HBR topics.

  1. Grammar: It is noted that your manuscript needs careful editing by someone with expertise in technical English editing paying particular attention to English grammar, spelling, and sentence structure so that the goals and results of the study are clear to the reader.

Answer: We have proofread the manuscript once more hoping that now it became clearer.

Reviewer 2 Report

In this study the authors report long-term clinical outcomes and prognostic factors in CHIP and HBR patients undergoing PCI with sirolimus-eluting stent implantation.

I have several concerns/questions.

-       Why patients died in hospital were excluded form the analysis, please explain. This should be clearly noted in the manuscript.

-       10 patients with cardiogenic shock at cath lab were excluded, however 6 other patients with cardiogenic shock were included? Please explain.

-       Age and gender should be included in the multivariate analysis.

-       CHIP was defined according to a publication by  Burzotta F. et .al, Int J Cardiol 2019, 293:84-90. What was the mean CHIP score of the patients included in this analysis.

-       You conclude that ‘PCI in CHIP and HBR patients is feasible with a low rate of periprocedural complications.  However, periprocedural complication e.g. thromboembolic, MI, no-reflow, dissection etc… are not mentioned in the results!.

na

Author Response

In this study the authors report long-term clinical outcomes and prognostic factors in CHIP and HBR patients undergoing PCI with sirolimus-eluting stent implantation.

I have several concerns/questions.

  1. Why patients died in hospital were excluded from the analysis, please explain. This should be clearly noted in the manuscript.

Answer: The paper aimed to assess the performance and safety of SES and it was all-comers study; therefore, we decided to exclude baseline deaths not related to the stent. We clearly stated this on the FlowChart (Figure 1) as well as in the main text. We made this as apparent as possible. “However, in 4 patients (5 stents), Alex Plus stents were not implanted (1 device failure – no possibility to deliver the stent to the target lesion due to calcification and tortuosity, 4 stents not implanted due to fatal cardiac arrest). Moreover, we excluded 14 patients (20 stents) due to in-hospital death unrelated to the sirolimus-eluting stent deployment.”

  1. 10 patients with cardiogenic shock at cath lab were excluded, however 6 other patients with cardiogenic shock were included? Please explain.

Answer: This is strictly associated with answer to question 1. If the patient was a priori in the cardiogenic shock (at admission), the stent implantation was a rescue/bail-out procedure, and the patient died on the cathlab table, then the patient was excluded (in this group no stent thrombosis was seen). However, if the patient was in the cardiogenic shock or develop in after admission, had implanted stent, and left the cathlab alive, then was included for further analysis. Making this FlowChart so frank we wanted to show how all-comers course looks like.

  1. Age and gender should be included in the multivariate analysis.

Answer: Sex and age were included in the univariable model, then, the multivariable Cox regression model was chosen in stepwise selection with a back-ward elimination algorithm with a significance level = 0.1. Please take a look at factors included in the univariable analyses – they are presented as Supplementary Tables S9 – S14.

  1. CHIP was defined according to a publication by Burzotta F. et .al, Int J Cardiol 2019, 293:84-90. What was the mean CHIP score of the patients included in this analysis. 

Answer: We included the following statement in the text: “The median number of met clinical criteria was 4 (IQR 2-7), and the median number of met anatomical criteria was 2 (IQR 1-3)”.

  1. You conclude that ‘PCI in CHIP and HBR patients is feasible with a low rate of periprocedural complications.  However, periprocedural complication e.g. thromboembolic, MI, no-reflow, dissection etc… are not mentioned in the results!

Answer: The exact data were presented in Table 3 and Supplementary Tables S5 and S6. However, now, we explored this issue in the text also. “The rate of coronary dissection was higher in CHIP patients than in non-CHIP (13.6% vs. 3.3%, p = 003) with similar rate of MI type 4a (1.2% vs. 2.7%, p = 0.66). There were no statically significant differences in periprocedural complications in HBR vs. non-HBR groups (dissection: 3.9% vs. 8.3%, p = 0.216 and MI type 4a: 1.3 vs. 2.6%, p = 0.999)”.